# Low *JAK2* V617F Allele Burden in Ph-Negative Chronic Myeloproliferative Neoplasms Is Associated with Additional *CALR* or *MPL* Gene Mutations

**DOI:** 10.3390/genes12040559

**Published:** 2021-04-12

**Authors:** Tatiana V. Makarik, Adhamjon O. Abdullaev, Elena E. Nikulina, Svetlana A. Treglazova, Elena E. Stepanova, Irina N. Subortseva, Alla M. Kovrigina, Anait L. Melikyan, Sergei M. Kulikov, Andrey B. Sudarikov

**Affiliations:** National Research Center for Hematology, Novy Zykovski lane 4a, 125167 Moscow, Russia; makarik_71@mail.ru (T.V.M.); adham_abdullaev@mail.ru (A.O.A.); lenysh2007@rambler.ru (E.E.N.); svetik997@yandex.ru (S.A.T.); stepanova173@gmail.com (E.E.S.); soubortseva@yandex.ru (I.N.S.); kovrigina.alla@gmail.com (A.M.K.); anoblood@mail.ru (A.L.M.); smkulikov@mail.ru (S.M.K.)

**Keywords:** myeloproliferative neoplasms, polycythemia vera, primary myelofibrosis, essential thrombocythemia, *JAK2* V617F, *MPL* W515L/K, *CALR*

## Abstract

*JAK2* (Janus kinase 2) V617F, *CALR* (Calreticulin) exon 9, and *MPL* (receptor for thrombopoietin) exon 10 mutations are associated with the vast majority of Ph-negative chronic myeloproliferative neoplasms (MPNs). These mutations affect sequential stages of proliferative signal transduction and therefore, after the emergence of one type of mutation, other types should not have any selective advantages for clonal expansion. However, simultaneous findings of these mutations have been reported by different investigators in up to 10% of MPN cases. Our study includes DNA samples from 1958 patients with clinical evidence of MPN, admitted to the National Research Center for Hematology for genetic analysis between 2016 and 2019. In 315 of 1402 cases (22.6%), *CALR* mutations were detected. In 23 of these 315 cases (7.3%), the JAK2 V617F mutation was found in addition to the *CALR* mutation. In 16 from 24 (69.6%) cases, with combined *CALR* and *JAK2* mutations, V617F allele burden was lower than 1%. A combination of *JAK2* V617F with *MPL* W515L/K was also observed in 1 out of 1348 cases, only. *JAK2* allele burden in this case was also lower than 1%. Additional mutations may coexist over the low background of JAK2 V617F allele. Therefore, in cases of detecting MPNs with a low allelic load *JAK2* V617F, it may be advisable to search for other molecular markers, primarily mutations in exon 9 of *CALR*. The load of the combined mutations measured at different time points may indicate that, at least in some cases, these mutations could be represented by different clones of malignant cells.

## 1. Introduction

Ph-negative myeloproliferative neoplasms (MPNs) arise as a result of the transformation of the hematopoietic stem cell, which leads to the aberrant clonal hematopoiesis and hyperplasia of the cells of the myeloid lineage [1]. The WHO (World Health Organization) classification of myeloid malignancies includes the following MPNs: Chronic myeloid leukemia (CML), chronic neutrophilic leukemia, polycythemia vera (PV), primary myelofibrosis (PMF), essential thrombocythemia (ET), chronic eosinophilic leukemia, and others, unclassified MPNs-U. These diseases are characterized by the hyperplasia of the myeloid cell lineages, thrombocytosis, splenomegaly, frequent thrombotic complications, with a long course—the risk of transformation to myelodysplastic syndrome and acute leukemia. Classical diagnosis, risk stratification, clinical characterization, and prognosis in patients with MPNs were based on histological analysis of bone marrow and peripheral blood. To date, it is known that the molecular mechanisms of MPN development are associated with hyperactivation of tyrosine kinases or with abnormalities of cytokine receptors. The beginning of deciphering these mechanisms started with the identification of the pathogenetic role of the chimeric *BCR/ABL* oncogene in chronic myeloid leukemia (CML), which arises as a result of reciprocal t(9:22) translocation forming the so-called Philadelphia chromosome [2,3]. The abnormal tyrosine kinase activity of the *BCR/ABL* protein leads to the dysregulation of many intracellular signaling pathways [4,5].

In 2005, a point mutation in exon 14 of the *JAK2* kinase gene was found in patients with MPNs, in which the amino acid valine was replaced by phenylalanine at position 617 (mutation *JAK2* V617F) in the JH2 pseudokinase domain of the *JAK2* protein [6,7,8,9]. The *JAK2* V617F mutation is found in 90–95% of PV patients, in 50–70% of ET cases, and 40–50% of myelofibrosis cases. The *JAK2* V617F mutation has proven to be a marker essential to performing the primary and differential diagnosis of MPNs [10,11]. Subsequently, 4% of *JAK2* V617F-negative PV patients were found to have a cluster of JAK2 gene mutations in exon 12 (microdeletions and point mutations affecting amino acid residues at position 537–542 of the JAK2 protein [6,7,8,9]).

In 2006, ET was found to associate with somatic point mutations in the gene for the thrombopoietin receptor *MPL*, causing amino acid substitutions W515L/K. These mutations are found in 5–10% of ET and PMF patients who do not have the *JAK2* V617F mutation and never with PV [12,13,14,15]. In vitro studies have shown that the replacement of tryptophan with leucine or lysine at position 515 of the *MPL* protein leads to the activation of the JAK2-STAT regulatory pathway, which also occurs in the case of the *JAK2* V617F mutation. In 2013, mutations in the calreticulin gene (*CALR*) were found in 67% to 88% of patients with ET and PMF [16,17]. To date, more than 40 different mutations have been found in exon 9 of the *CALR* gene. All of them are insertions and/or deletions that lead to the formation of a new C-terminal protein sequence and the loss of the KDEL signal sequence due to a shift in the reading frame [16,17,18]. The most common mutations are 52 base pair deletion (type 1) and 5 base pair insertion (type 2) [17]. Mutant forms of calreticulin can bind to the extracellular domain of *MPL* and, thus, induce the activation of the JAK2-STAT/P13K/MAPK signaling pathway, which leads to increased cell proliferation. Some current studies report lower *CALR* mutation rates than those reported in earlier studies, and it is believed that CALR mutation rates may vary by race and country [16,17,19,20]. The overall survival of patients with *CALR* type 2 and *JAK2* mutations is significantly lower than the survival rate of patients with type 1 mutations [21,22]. Basically, the molecular diagnostic routine for ph-negative MPNs starts with the test for *JAK2* V617F and does not proceed further since the V617F positivity obtained. Therefore, studies of *CALR* and *MPL* mutations are rarely performed on patients with *JAK2* mutation, and the data on the mutation coexistence are limited [13,14,20,23,24,25,26]. A systematic study of patients with both mutations and the combined effect on the pathophysiology, treatment outcome, phenotype, and clinical characteristics of patients with MPN is required [12].

The aim of this study was to assess the frequency of detection of *JAK2* V617F, *CALR,* and *MPL* mutations, in particular in cases of combined *JAK2* + *CALR*, *JAK2* + *MPL* mutations, in a Russian cohort of patients with *BCR/ABL1* rearrangement negative MPNs and factors associated with them.

## 2. Materials and Methods

A retrospective study of patients observed at the National Research Center for Hematology (Moscow, Russia) from October 2016 to November 2020 was carried out. All subjects gave their informed consent for inclusion before they participated in the study. The study was conducted in accordance with the Declaration of Helsinki, and the protocol was approved by the Ethics Committee of National Research Center for Hematology (Protocol # 153). All additional tests were performed on the material left after all the necessary diagnostic tests were performed and the correct diagnosis was made for 1958 patients included in the study. Data on gender, age, and diagnosis of patients were taken from medical records at the time of diagnosis of ET, PV, or PMF during outpatient consultation. The diagnosis was made taking into account the effective WHO recommendations (2008 or 2016) [27,28].

Determination of possible genetic mutations in *JAK2* V617F, exon 9 of *CALR* (types 1 and 2), and *MPL* W515L/K, as well as mutations of 12 exon *JAK2*, was performed. When negative results were obtained for the above mutations, a “triple-negative” status was established.

DNA and RNA were isolated out from 5–10 mL of blood or bone marrow, taken at the time of diagnosis, using standard “salting out” procedure or by the “Ribo-sol-D” kit (Interlabservice, Moscow, Russia) following the manufacturer’s instructions. The concentration and purity of DNAs were assessed spectrophotometrically.

The *JAK2* V617F mutation was quantified using the “*JAK2* Gene V617F G/T Mutation Detection Kit” (NP_404_Q_RG, Syntol, Moscow, Russia) according to the manufacturer’s instructions. *MPL* W515L/K mutations were assessed using the “Reagent kit for determining the W515L/K mutation of the *MPL* gene” (NP_412_100, Syntol, Moscow, Russia). The sensitivity of allele specific PCR assays was 0.2% of mutant allele over wild type allele background according to the manufacturer’s instructions. For PCR amplification, a RotorGene device (QIAGEN, Hombrechtikon, Switzerland) was used.

A pair of specific oligonucleotide primers were used to amplify the 9th exon of the *CALR* gene: CALR_AF1 5′-CTGAGGTGTGTGCTCTGCC-3′ and CALR_ R FAM 5′-CAGAGACATTATTTGGCGCGG-3′. The reaction mixture contained: 2.5× reaction mixture (Syntol, Moscow, Russia), MgCl_2_ 25 mM, 10 pmol primers, and the final volume of the mixture was 25 μL (for the amplification, a Bio-Rad device, Hercules, CA, USA was used). The amplification program included: Denaturation 95 °С/10 min; cycling 35 cycles: 95 °C/20 s, 60 °C/40 s, 72 °C/60 s, and 72 °C/10 min. PCR products were diluted with water 80 times, denatured at 95 °C/3 min, and cooled at 4 °C/5 min. Fragment analysis with a sensitivity of at least 3% was performed in a NANOFOR-05 genetic analyzer (Syntol, Moscow, Russia).

*JAK2* exon 12 mutations were evaluated by direct Sanger sequencing of PCR amplified exon 12. The sensitivity of detection was ~15% of mutated allele.

All cases were tested for ph-negativity using the Reverta-L and AmpliSense FRT variant kits (Interlabservice, Moscow, Russia) following the manufacturer’s instructions.

For statistical data processing, the classical methods of descriptive, frequency analysis, and logistic regression were applied. We used SAS 9.4 software for all calculations [29]. Receiver Operating Characteristic (ROC) analysis was performed using SAS Macro %ROC-CUTOFF, written by K. Harris [30].

## 3. Results

In total, a combined group of patients (1958) with diagnoses of ET, PV, PMF, and MPN-U was examined for the presence of *JAK2* V617F, *JAK2* exon 12, *MPL* W515L/K, and *CALR* exon 9 gene mutations.

To analyze the occurrence of mutations, including combined ones, we formed two age groups. The first group consisted of patients under the age of 60, the second—older patients (the ratio of age groups was 29.4% [576/1958] versus 70.6% [1382/1958], *p* = 0.037). There were 2 times more women (1289) in the general group than men (670), probably due the predominance of ET that is more common in woman, than in man [31]. The distributions of mutation detection frequencies in gender, age, and diagnosis groups are shown in Table 1.

According to our data, 68.8% (1913/1958) of patients with MPNs had the *JAK2* V617F mutation. The overall frequency of mutations in the *CALR* gene was 22.6% (316/1401). *MPL* mutations were much less common: 1% *MPL* W515L (14/1354) and 0.7% *MPL* W515K (9/1350). For mutations in exon 12 *JAK2*, the frequency of occurrence was found to be 4.7% (5) in 107 patients with MPNs. Our results are comparable with previously published data: *JAK2* V617F—62% [12], *CALR*—15.5% [28], and *MPL*—5% [12,17].

In the group of patients with PV, the *JAK2* V617F mutation was detected in 91.1% (509) of patients. In this group of patients, not a single case with the *MPL* W515L/K or *CALR* mutation was found, while a mutation in exon 12 JAK2 was detected in five cases (8.9%) and only in patients with PV.

The detection rate of *MPL* W515L/K was slightly higher in the group with PMF—3.4% (7) versus 1.5% (4) in the group with ET (*p*χ2 = 0.03).

There was no significant difference between the mutation frequencies observed in males and females for any of the MPN patient cohorts.

The frequency of *JAK2* V617F mutations in patients with PMF and ET was 60.5% and 53.9%, respectively (Table 1). It is known that the *JAK2* V617F mutation is considered as one of the factors that increase the risk of thrombosis and is included by WHO as a criterion in the IPSET-thrombosis scale for all MPNs [23]. Patients with MPN-U had the *JAK2* V617F mutation in 61.9% of cases.

The incidence of mutations in exon 9 of *CALR* in patients with ET and PMF was 40.3% and 36.9%, respectively. Mutations in the *CALR* gene were significantly more frequent (*p* = 0.0006) in younger patients, 27.7% versus 20.2%.

Mutations in the *MPL* gene were found in 3.4% of patients with PMF, and 1.5% of patients with ET. In the older group of patients, mutations in the *MPL* gene were much more common (the ratio of age subgroups 2.9% [13/450] versus 1.1% [10/914], *p* = 0.02).

A mutation in exon 12 *JAK2* was found only in patients with PV in 8.9% of cases (*p* = 0.19), which does not contradict previously published data [6,7,8,9].

In addition to the frequency estimates, we also checked possible differences in the distributions of the *JAK2* and *CALR* allelic load in the groups by gender, age, and diagnosis (Table 2 and Figure 1).

According to the results of our study, there were no significant differences in the CALR allelic load distributions by age groups and gender.

In the case of the *JAK2* V617F mutation, we found significant differences in terms of age: In patients over 60 years of age, the allelic load is higher than in younger patients (with a median value of 40% [SD 1.5] versus 20% [SD 0.9]) (Figure 2).

As a result of our study, 24 cases were found with the simultaneous presence of combined mutations (CM) *JAK2* V617F/*CALR* and *JAK2* V617F/*MPL* W515L/K. All of these results were confirmed at two or more time points. The combination of *JAK2* V617F mutations and exon 9 of the *CALR* gene occurred in 23 cases, and the combination of *JAK2* V617F and *MPL* W515L/K mutations—1 case. Clinical and laboratory data of MPN patients with CM are presented in Table 3. The information was available for 23 patients. In one case, after being diagnosed, the patient was followed up and treated at another clinic. We analyzed the data of the entire group of patients, and separately for patients with ET and PMF. Due to the small number of observations (3 cases), the group of patients with MPN-U were not separately characterized.

Type 1 *CALR* mutation was found in 16 patients (69.6%) and type 2 mutation was found in 7 patients (30.4%).

When analyzing combined mutations, we found that 70% of all values of the *JAK2* V617F allelic load were not significantly higher than 1% (minimum value 1%, maximum value—97%, st. dev. 4.2), while the median values of allelic load in the general group—37.5% (minimum value 10%, maximum value 53%, st. dev. 5.2) (Figure 3).

To study this phenomenon, ROC analysis was carried out to assess if the level of *JAK2* mutation allele burden can be hypothetically used as an indicator test to predict the presence of additional mutations.

As shown by ROC analysis, the level of *JAK2* mutation allele burden is a good test indicator for predicting the appearance of additional *CALR* mutations (area under the ROC curve, AUC = 0.86, Figure 4). The ROC analysis for *JAK2* and *MPL* co-mutations was not possible due to only one such case noted.

Furthermore, we proceeded from practical considerations that a high level of specificity, in this case, is more important than sensitivity, i.e., we would like to suspect the presence of additional mutations with a reasonable degree of certainty. For this, we have chosen as a threshold the minimum detectable level of allelic load = 1%.

Based on this rule, we classified the sample (observations with a known measured level of *JAK2* V617F allele burden) into two groups with predictable additional *CALR* mutation detection and without it and compared it with the actual presence. The cross tabulated frequencies are shown in Table 4.

As can be seen from Table 4, the total number of observations that do not satisfy the proposed rule are 25 (18 + 7) out of 794 (3.1%). In 23 patients with combined mutations, *JAK2* V617F allele burden equal or lower than 1% was found in 16 patients (69.6%). The chance to detect CALR mutation in the case of a low *JAK2* V617F allele burden is about 96 times higher than in the case of a high burden (OR = 95.6 (35.0–260.9)). Other statistics describing the classification rule (allele burden <>1%) follow: Coefficient of agreement Kappa = 0.55 (0.39–0.70), *p* Fisher <0.0001.

The analysis of the *CALR* mutation load as a possible indicator of the presence of combined mutations did not provide any practical prognostic rule. The area under the curve for the “*CALR* indicator” AUC = 0.54, the levels of *CALR* loads in patients with an isolated CALR mutation, and patients with combined *CALR-JAK2* did not practically differ.

For 13 out of 24 patients with combined mutations, multiple archival samples taken at different time points were available. Eight patients had a stable burden for both *JAK2* and *CALR* mutation during follow up. On the other hand, five patients significantly dropped the level of *JAK2* mutation under therapy while preserving the *CALR* mutation load. In one case, the decrease in *JAK2* V617F allele burden was from 43% to negativity; in one case from 49% to 1% and in three cases from 1% to negativity. Therefore, one can speculate that some of the cases with combined mutations are representing at least two-clonal diseases.

## 4. Discussion

Currently, the pathogenetic role of mutations occurring separately *JAK2* V617F, 12th exon *JAK2*, *MPL* W515L/K, and *CALR* exon 9 in MPNs is well understood. However, in cases of a combination of these mutations in patients with MPNs, their significance for pathogenesis and, as a consequence, the expected effectiveness of therapy (including targeted therapy) requires further study. Since the common workflow for molecular diagnostics in ph-negative MPNs, which suggest *CALR* and *MPL* mutation testing in *JAK2* V617F negative cases only, there are few data obtained on limited cohorts concerning a combination of *JAK2* V617F, *CALR*, and *MPL* point mutations available so far. Previously, Lim et al. reported detection by high-resolution melting analysis of both classic and non-classic CALR exon 9 mutations on a cohort of 59 *JAK2* V617F-mutated ET patients [26]. Here we presented the data on the retrospective search for the combined mutations on the extended patient sample (*n* = 1958), including all patients that were admitted to the National Research Center for Hematology (Moscow, Russia) from 2016 till 2020. We showed that CM (namely *JAK2* V617F and *CALR* mutations or *JAK2* V617F and *MPL* W515L/K) could occur not only in patients with ET but also in patients with PMF and MPN-U. We analyzed the type and allelic load of these mutations in 24 patients with CM and have shown for the first time that in most cases, additional mutations exist over the background of *JAK2* V617F mutation with a low allelic load. In 70% of cases (16/24), the *JAK2* V617F allele load was ≤1%. Therefore, in case of detecting MPN with a low allelic load of *JAK2* V617F mutation, it may be advisable to search for other molecular markers, primarily mutations in exon 9 of *CALR*. Dynamic observation of *JAK2* V617F and *CALR* mutation burden in combined cases indicated that these mutations may exist in different malignant cell clones. Unfortunately, only archival DNA samples were available for retrospective analysis, therefore cell separation to prove the oligoclonal nature of the disease was not possible.

## 5. Conclusions

Patients with a low JAK2 V617F allele level may harbor additional mutations in the CALR or MPL genes. Therefore, in cases of MPNs with a low allelic load of JAK2 V617F, it may be advisable to search for other molecular markers, primarily mutations in exon 9 of CALR. Some cases with the level of combined mutations being measured at different time points indicate different dynamics of mutations under the influence of therapy. It can be assumed that, at least in some cases, these mutations can be represented by different clones of malignant cells. To assess the role of additional mutations in the prognosis of MPNs and to determine the tactics of therapy (including targeted), further identification, monitoring, and prospective study of such patients are necessary.

## Figures and Tables

**Figure 1 genes-12-00559-f001:**
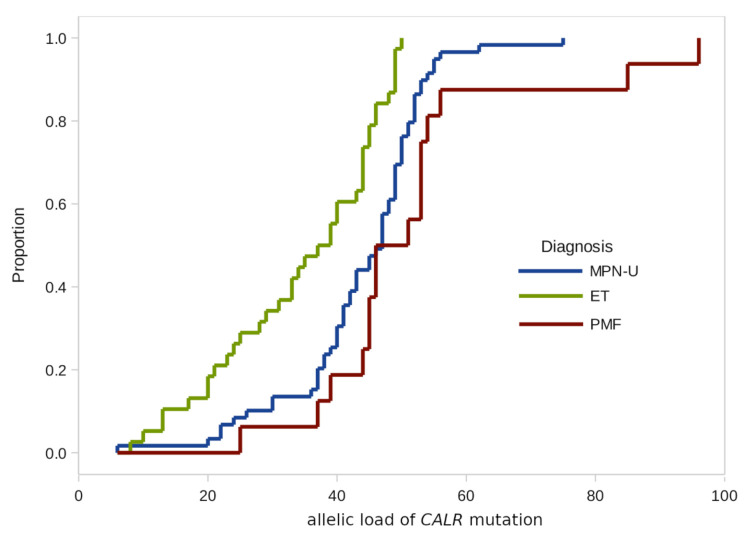
Empirical distributions of *CALR* mutation burden in patients with different diagnoses.

**Figure 2 genes-12-00559-f002:**
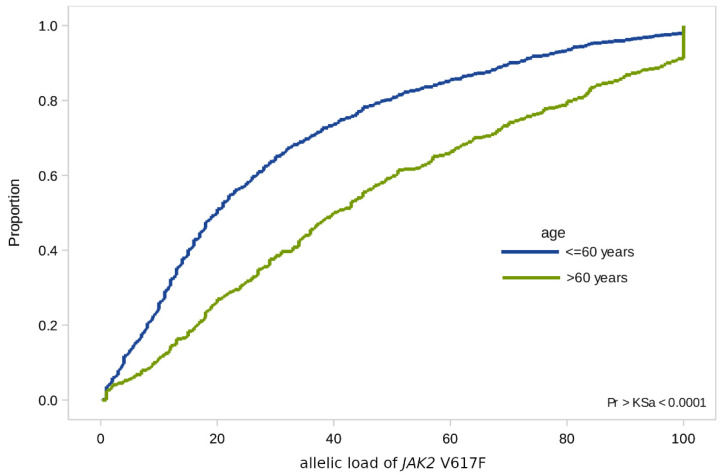
Empirical distributions of *JAK2* V617F allele load in patients of two age groups: Under 60 and over 60.

**Figure 3 genes-12-00559-f003:**
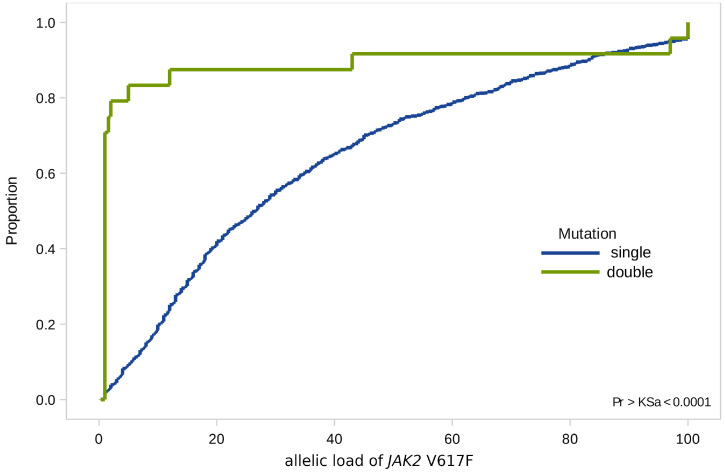
Empirical distributions of *JAK2* V617F allele load in patients with single (*JAK2* V617F) and double (*JAK2* V617F and *CALR*) mutations.

**Figure 4 genes-12-00559-f004:**
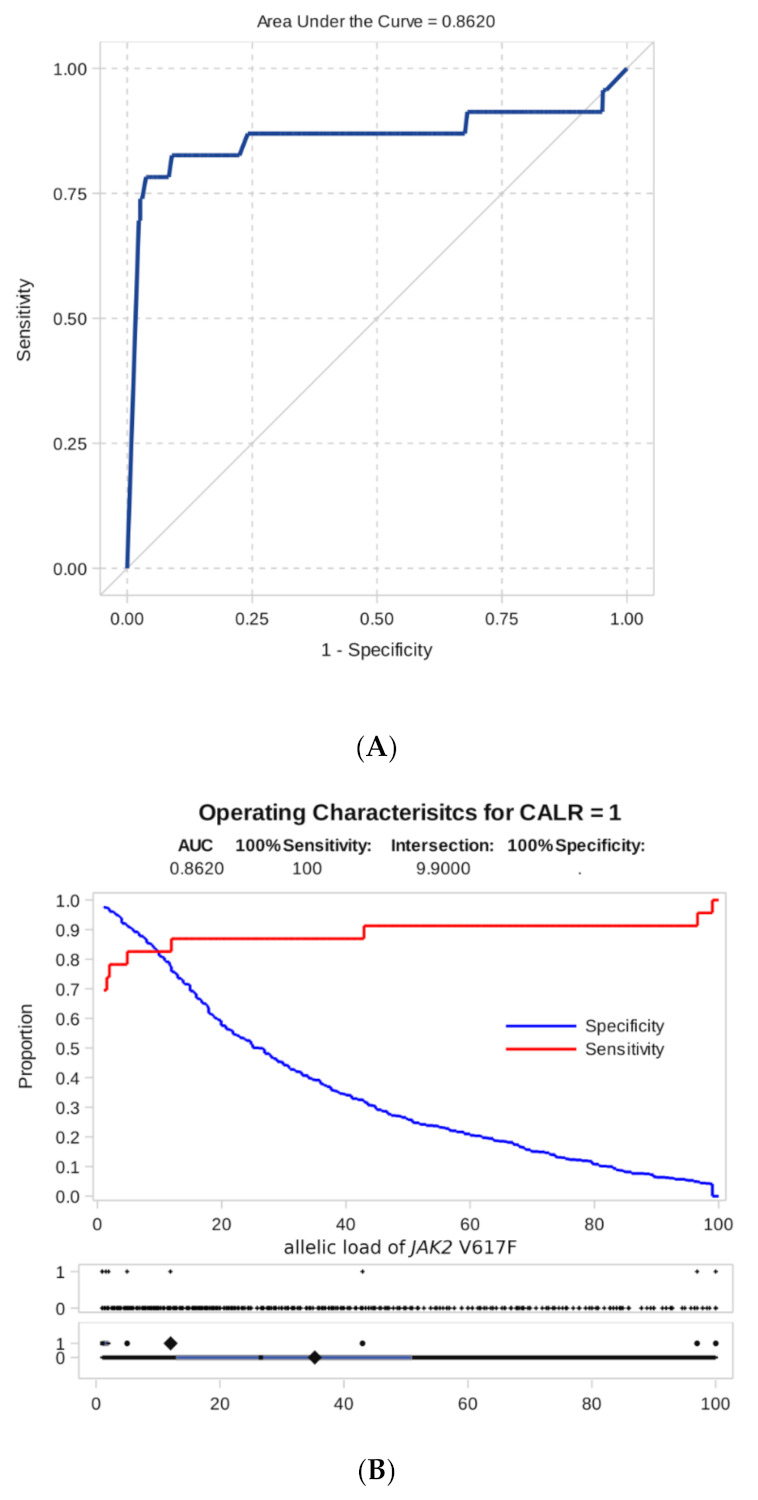
Result of ROC analysis. (**A**) ROC curve of dependence of sensitivity on specificity for an indicator of predicting the presence of a combined mutation according to the level of the *JAK2* V617F allele burden. (**B**) Sensitivity and specificity of detecting a combined mutation as a function of the threshold level of the *JAK2* V617F allele burden.

**Table 1 genes-12-00559-t001:** General characteristics of the study cohort of patients. *p*-value from Fisher’s test was used to estimate significance for the tables of 2 × 2 and χ-square for the tables of higher dimensions (2 × 3, 2 × 4).

Index		Mutation Frequency	ODR (Odds Ratio)/*p*
*JAK2*+			
In cohort		68.8% (1914/1958)	
Gender	Male	67.4% (443/657)	ODR = 1.1 (0.9 − 1.35) *p*Fisher = 0.2
	Female	69.5% (874/1257)	
Age	≤60 years	66.2% (339/559)	ODR = 1.5 (1.2 − 1.9) *p*Fisher = 0.0001
	>60 years	74.6% (446/598)	
Diagnosis	ET	53.9% (185/343)	*p*χ2 < 0.0001
	PV	91.1% (507/555)	
	PMF	60.5% (156/258)	
	MPN-U	61.9% (469/758)	
*CALR*+			
In cohort		22.6% (315/1402)	
Gender	Male	21.0% (102/485)	ODR = 1.1 (0.9 − 1.5) *p*Fisher = 0.2
	Female	23.2% (213/917)	
Age	≤60 years	27.7% (118/426)	ODR = 0.6 (0.4 − 0.8) *p*Fisher < 0.0002
	>60 years	20.2 (197/975)	
Diagnosis	ET	41.2% (112/272)	*p*χ2 < 0.0001
	PV	0	
	PMF	36.9% (79/214)	
	MPN-U	19.8% (124/628)	
*MPL*+			
In cohort		1.7% (23/1364)	
Gender	Male	1.7% (8/474)	ODR = 0.9 (0.4 − 2.4) *p*Fisher = 0.6
	Female	1.7% (15/890)	
Age	≤60 years	1.1% (10/914)	ODR = 2.7 (1.2 − 6.2) *p*Fisher = 0.02
	>60 years	2.9% (13/450)	
Diagnosis	ET	1.5% (4/263)	*p*χ2 = 0.03
	PV	0	
	PMF	3.4% (7/205)	
	MPN-U	1.9% (12/609)	

**Table 2 genes-12-00559-t002:** Statistical parameters of *JAK2*+ and *CALR*+ allelic loads by main demographic factors. Comparison of means by one-way analysis of variance (ANOVA), *p*-value from Fisher’s test was used to estimate significance.

Factor		N	Median, Range	Mean	Standard Deviation	*p* Fisher
*JAK2*+						
Gender	Male	438	29.0 (0.28–100)	37.1	28.6	
	Female	859	24.0 (1–100)	33.8	28.9	*p* = 0.05
Age	≤60 years	856	20.0 (0.28–100)	28.9	25.7	
	>60 years	441	40.0 (1–100)	46.5	30.9	*p* < 0.0001
Diagnosis	ET	182	10.8 (1–100)	14.4	13.7	
	PV	497	34.0 (1–100)	41.3	28.6	
	PMF	154	43.0 (1–100)	47.2	30.7	
	MPN-U	464	22 (0.28–100)	32.2	28.1	*p* < 0.0001
*CALR*+						
Gender	Male	40	42.0 (10–54)	38.7	12.9	
	Female	73	45.0 (6–96)	43.4	14.1	*p* = 0.08
Age	≤60 years	89	43.0 (6–96)	41.5	12.9	
	>60 years	24	46.0 (8–85)	42.3	16.7	*p* = 0.7
Diagnosis	ET	38	38.0 (8–50)	34.5	12.1	
	PV	0				
	PMF	16	48.5 (25–96)	51.8	17.1	
	MPN-U	59	47.0 (6–75)	43.8	11.1	*p* < 0.0001

Combined mutations.

**Table 3 genes-12-00559-t003:** Clinical and laboratory characteristics of MPN patients with combined mutations.

Criteria	Whole Group (*n* = 23)	ET (6)	PMF (14)
Age, years, median (range)	57 (23–86)	46 (26–61)	64.5 (23–86)
Men:Women	1:1 (12:11)	1:1 (3:3)	5:9 (1:1.8)
Diagnosis: ET PMF MPN-U	6 14 3		
Red blood cell, ×10^12^/L	4.1 (2.9–5.7)	4.4 (3.5–5.1)	3.89 (2.9–4.8)
Hemoglobin, g/L	120 (83–166)	133 (83–154)	106 (91–142)
Platelets, ×10^9^/L	815 (98–1900)	1298 (774–1900)	572 (98–1475)
White blood cell, ×10^9^/L	7.6 (3.41–16)	8.1 (7.43–16)	6.6 (3.41–11.63)
Splenomegaly, % (number of patients)	61% (14)	17 (1)	79 (11)
Risk:			
IPSS Low Intermediate-1 Intermediate-2 High		NA	4 7 3 0
IPSET: Low Intermediate High		0 4 2	NA
Thrombosis, % (number of patients)	9% (2)	33% (2)	-
Treatment:			
Hydroxyurea	16	4	10
Interferon α 2b	2	1	0
Anagrelid	1	1	0
Imatinib then Interferon α 2b	2	0	2
Observation	2	0	2

**Table 4 genes-12-00559-t004:** Association of low *JAK2* allele burden with *CALR* mutation. *p*-value from Fisher’s test was used to estimate significance.

	*CALR*+	*CALR*−	Total
Allele burden *JAK2* ≤ 1%	16	18	34
Allele burden *JAK2* > 1%	7	753	760
Total	23	771	794

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
