# Peer review of "Low JAK2 V617F Allele Burden in Ph-Negative Chronic Myeloproliferative Neoplasms Is Associated with Additional CALR or MPL Gene Mutations"

_genes, 2021, doi:10.3390/genes12040559_

Round 1

Reviewer 1 Report

The authors present a retrospective study of 1958 patients with MPN and focus on the co-occurrence of JAK2 mutations with CALR or MPL mutations. The article's strength lies in the large series of patients analyzed. Also, the conclusion that MPN patients found to have a low JAK2V617F VAF should be analyzed for the presence of a CALR co-mutation is very interesting. However, the study lacks novelty and gives no new information on the clinical characteristics of co-mutated patients. Also, the article is confusing in parts.  
  1. Introduction
  • Line 54 - the authors mention the monitoring of MRD. This is not current practice in MPN patients. 
  • Line 73 - the authors state that patients with a CALR type 2 mutation have 4 points in DIPSS, but neither DIPSS nor DIPSS-Plus given points for CALR mutation. And at higher risk of what?
  • Line 81 - suggest inclusion of reference Lim KH et al. Blood Cancer J. 2015. They analyzed 92 MPN patients and identified 16/59 JAK2 mutated patients as co-mutated in CALR.
  2. Methods
  • Line 91 - It's impossible that the WHO 2016 recommendations were used for patients diagnosed in October 2014 
  • Line 96 - "9 exons" should be exon 9
  • What technique was used to detect JAK2 exon 12 mutations?
  • There is no mention of the sensitivity of the techniques used. What is the limit of detection? 
3. Results
  • Line 128 - why was the cohort split into an under 40 group and over 40? In general, patients with MPN aged under 60 could be considered "young".
  • Line 147-148 is very confusing.
  • In Table 1, and in the article in general, there is a lack of clinical characteristics.
  • As above point, this is especially important for the PV patients found to have co-mutated JAK2 and CALR. The authors should add these patient's clinical characteristics and specific details of the mutations identified in a supplementary table. This is particularly important so that readers can confirm that these were not "overt PV" or misdiagnosed ET patients.
  • Table 2 has a mix of 0,05 and 0.08 decimal usage.
  • Figure 4 is only for CALR mutation. Was a similar analysis performed for JAK2 and MPL co-mutations?
  • Line 234-237 is confusing. Not sure how many patients they are talking about: 25, 23, 16? This needs re-writing as the message is lost.
  • Line 235: "extremely low" VAF - what is the value? The authors only mention less than 1%, but never state the absolute values. Again, the LOD of the methods is not stated so it is impossible to know what values they are talking about, and with what sensitivity the technique has.
4. Discussion
  • Line 255. The authors mention "therapy." What therapy? For what MPN? This is also mentioned in results line 248, but very vague.
  • As above, the authors mention that the effect of co-mutation of therapy should be investigated, but offer no data on their cohort in relation to therapy received. Could they not look at this? This would add novelty to their article.
  • Use of MPN, NPM (line 257), MPD (line 53) is inconsistent.
  • The authors should give more discussion on the detection of CALR mutations in 4 PV patient as there are very little published on this. Due to its novelty this result is interesting/significant and should be mentioned in the abstract.
  • The conclusion on line 268 is very interesting and fully supported by the authors' results. I would definitely suggest re-writing of the abstract to include this observation.

Reviewer 2 Report

Authors have assessed the co-occurrence of CALR, MPL and JAK2 V617F or exon 12 mutations in a cohort of 1958 (or 1959?) patients.

Major

The material used in the study is unclear. Blood and bone marrow DNA from MPN it is stated on line 100. What is used? And if it varied, this should be stated as VAF will vary depending on source. Was the material collected at MPN diagnosis or are these cross-sectional samples and some patients may be under treatment?

What are the detection limits of the assays applied? Are indeed the PCR false positive at the low vaf of 1%?  What housekeeping genes are run with the pcr? How is the reproducibility?

Have the findings been cross referenced e. g to clinical data on mutations or other method to validate with another method that these are true mutations and not PCR errors?

JAK2 and CALR were found in 4 patients with PV, were these really PV? What do you mean by the sentence that the diagnostic criteria can vary between diagnosis. These 4 need to be re-examined, are they really PV? Line 154-160

Figure 3 and 4.  What is the relevance of this?  And again, are these “real” mutations, what is the detection limit of the PCR assay and what were the housekeeping genes?

Minor

The manuscript contains no info on the distribution of CALR type 1 and 2 among the coexisting CALR with JAK2. Please supply this information.

It is stated 1958 and 1959 patients in different places. Line 91 e.g. What is the correct number? also for CALR, is it 23 or 24? different statements at different lines.

In abstract 10/% why do you use / on line 14

Overall: in figures and table when “MPN” is used, you probably mean MPN-U. This should be corrected.

1959 “clinical MPN” what does this mean. And is it 1958 or 1959? And mutation analyses seems not to be performed in all, CALR in 1401 of 1958 and MPL in 1363? Why did you not perform the analysis on all?

Line 130 women were 2 times men. Please correct language and also comment on this gender distribution.

Line 147 gender peculiarities in the structure. What do you mean?

Line 152 Unencified? This is not a word

Table 2. Fisher p test ET PV PMF against what? And MPN, do you mean MPN-U? please state this.

Line 234 Erroreously? What do you mean?

Figure 1. Empirical distribution. Please rephrase. MPN is MPN-U? if so, state this.

Table 4. Table text needs elaboration

Round 2

Reviewer 1 Report

The authors have addressed all the points raised in the first round of peer review. The paper is definitely improved as a result. However, before the manuscript can be accepted for publication there are still some minor points that should be addressed.

  • The abstract is better, but please remove the phrase "The load.... of malignant cells." since the authors present no data to support this statement.
  • Line 134, include PV.
  • Table 1, there is still a comma (67,4%)
  • Line 152: 91.1% or 91.2%. Please check the value cited in the text corresponds to the value in the table.
  • Line 153. Mention that not a single MPL or CALR mutation was found.
  • Line 158. This phrase is still confusing. Perhaps this could be reworded to "There was no significant difference between the mutation frequencies observed in males and females for any of the MPN patient cohorts."
  • In relation to the above statement, in Table 2 it appear that the % of JAK2 mutation was p=0.05 for males vs females. Does this not contradict this observation?
  • The authors have included clinical characteristics of 23 CM patients in table 3. But what about the whole cohort? It is important that the reader can compare for example the clinical characteristics of an ET patient with a JAK2 mutation or CALR mutation with those of an ET CM patient. Please include these in a supplementary table.
  • Line 177, remove this phrase since no PV patients had CALR CM.
  • The figures would benefit from some formatting.
  • Gene usage/nomenclature. When referring to mutation in the JAK2 gene, the gene should be in capitals and italics. This should be consistent throughout the text, tables and figure legends.
  • The discussion is short and simply reiterates the key results. No references are cited. The authors should reflect on their results in relation to the published literature, comparing and contrasting what is previously published. For example, the observation that the risk of CM increases with age, has this previously been reported? The observation that the majority of patients with CM had the JAK2 mutation with low VAF, has this previously been observed in other cohorts that report MPN patients with co-mutations? 

Reviewer 2 Report

The authors have now clarified most queries, however minor corrections especially with regard to English language still need to be carried out. 

Abstract 

The last 4 lines have been modified however the last sentence should be rephrased “may indicate that, at least in some cases, these mutations may be represent…” as the authors have not produced any evidence of several separate clones.

Line 140-142 is still not correct English wording and needs to be rephrased and corrected. Line 142 change sex-age and diagnosis groups (to gender, age and diagnosis)

Line 158-160 Analysis of… this needs rephrasing, what do you mean, bias in structure?

Please clarify line 317-320 “the table” needs to be Table 4. Please provide an explanation of the statistical calculation, what is calculated? What is Kappa? And what is Px2<0.028?
